# Direct electrical stimulation of the premotor cortex shuts down awareness of voluntary actions

Luca Fornia[1,6], Guglielmo Puglisi [2,3,6], Antonella Leonetti[1,3], Lorenzo Bello[3], Anna Berti[4,5], Gabriella Cerri[1] & Francesca Garbarini [2,5]*

A challenge for neuroscience is to understand the conscious and unconscious processes underlying construction of willed actions. We investigated the neural substrate of human motor awareness during awake brain surgery. In a first experiment, awake patients performed a voluntary hand motor task and verbally monitored their real-time performance, while different brain areas were transiently impaired by direct electrical stimulation (DES). In a second experiment, awake patients retrospectively reported their motor performance after DES. Based on anatomo-clinical evidence from motor awareness disorders following brain damage, the premotor cortex (PMC) was selected as a target area and the primary somatosensory cortex (S1) as a control area. In both experiments, DES on both PMC and S1 interrupted movement execution, but only DES on PMC dramatically altered the patients' motor awareness, making them unconscious of the motor arrest. These findings endorse PMC as a crucial hub in the anatomo-functional network of human motor awareness.

[1] Laboratory of Motor Control, Department of Medical Biotechnologies and Translational Medicine, Università degli Studi di Milano, Humanitas Research Hospital, IRCCS, 20089 Milano, Italy. [2] MANIBUS—Movement and Body in Behavioral and Physiological Neuroscience—Lab, Psychology Department, University of Turin, 10124 Turin, Italy. [3] Neurosurgical Oncology Unit, Department of Oncology and Hemato-Oncology, Università degli Studi di Milano Humanitas Research Hospital, IRCCS, 20089 Milano, Italy. [4] SAMBA—SpAtial, Motor & Bodily Awareness—Research Group, Psychology Department, University of Turin, 10124 Turin, Italy. [5] Neuroscience Institute of Turin (NIT), 10124 Turin, Italy. [6]These authors contributed equally: Luca Fornia, Guglielmo Puglisi. *email: francesca.garbarini@unito.it

Willed actions are generated through a chain of unconscious events, although we are usually aware of moving (or not moving) our body according to a desired state. How does this motor awareness emerge from our brain? A critical contribution for understanding the anatomical substrate of human motor awareness comes from neuropsychological disorders following brain lesions, such as anosognosia for hemiplegia (AHP), where patients are firmly convinced of actually executing voluntary movements with their paralyzed limb[1–6]. According to an anatomo-clinical model of AHP[7–9], lesions to the premotor cortex (PMC) and neighboring areas impair patients' motor awareness by preventing the detection of the mismatch between intended, but not executed, movements with the paralyzed limb. Hence, PMC has been proposed as a crucial player in a comparator system[10,11] that has to match expected and actual motor outputs, in order to achieve conscious monitoring of voluntary actions.

In the present study, we test the role of PMC as a shared neural substrate for motor execution and motor awareness of voluntary actions, by using DES in patients with a left hemisphere brain tumor undergoing awake surgery. To this aim, 12 awake patients perform a voluntary motor task with their right hand, consisting of rhythmical manipulation of a cylindrical handle (hand-manipulation task (HMt), see Fig. 1)[12]. In eight patients, the HMt is coupled with an online verbal motor-monitoring task (MMt), in which the patient have to evaluate in real time his/her motor performance, stating OK when the action is executed without experiencing any difficulty, and stating STOP when some difficulties is noticed. While patients perform both tasks, target (PMC, ventro-lateral BA6—in four patients) and control (hand-fingers primary somatosensory cortex, S1—in other four patients) brain areas are transiently disrupted by using a low frequency (LF)-DES protocol[13]. To face problems of anarthria and dysarthria, possibly occurring during PMC stimulation and affecting the reliability of the online task (see below), in four additional patients, the HMt is coupled with a delayed MMt, in which patients has to report their motor performance immediately after DES. See details in Methods.

As reported in a previous study using the same motor task, DES on PMC interferes with motor execution[12,14,15]. Does the disruption of PMC simultaneously interfere with the patients' motor awareness? We anticipate that, if the PMC acts as a comparator system, an erroneous report during the MMt (i.e. impaired awareness) is expected as a consequence of transient disruption of PMC activity. In the control condition, when the same procedure is applied to S1, we expect DES interference with motor execution (altered HMt) (ref. [12], see also ref. [16]) but not with motor awareness (unaltered MMt). Indeed, in normal conditions, it has been demonstrated that motor awareness can be independent from somatosensory feedback. In other words, motor awareness is not retrospectively constructed on visual or proprioceptive information coming from the moving muscles but is based on a prediction model triggered by the subject's motor intention (e.g. ref. [17]). In Fourneret and Jeannerod's words[18] we

are aware of the movement that we intend to perform and not of the movement that we have actually performed. Furthermore, in a pathological context, double dissociations between AHP and both tactile and proprioceptive deficits have been described, suggesting that patients with spared somatosensory feedback can show impaired motor awareness[7].

In agreement with our predictions, the present findings show that transient disruption of the PMC during voluntary movement not only causes a motor arrest, but makes the patients unconscious of the induced block of action execution. The role of the PMC as a shared neural substrate for both motor execution and motor awareness is discussed.

## Results

**Effect of DES on the HMt.** DES (intensities range: 2–4.5 mA) applied on both PMC (in eight patients) and S1 (in four patients) produced a clear motor impairment (i.e. the hand movements and the task were completely abolished) in 27 out of 47 stimulated sites (17 over PMC and 10 over S1), as detected by both electromyography activity of hand muscles (Fig. 2e) and video-recording of behavioral outcome (see Supplementary Movies 1 and 2). The effect of DES on both PMC (Fig. 2b, c; white and red dots) and S1 (Fig. 2a) evoked suppression of the activity in all muscles considered. In all trials, movements restarted after stimulation. Although the movement interruption during DES was easily identified, an electromyography (EMG) signal analysis was performed in order to quantify the effect of DES on the hand muscles during HMt execution. The results showed that, in both S1 and PMC, DES during HMt execution induced a muscle suppression with respect to baseline (for PMC $H = 43.29$, $p = 0.000$; for S1 $H = 28.38$, $p = 0.000$; Fig. 2e). Additionally, during the hand-rest condition, DES on PMC and S1 did not evoke any observable movement or EMG activation. See Fig. 3 for detailed HMt results of each stimulation site in each patient.

**Effect of DES on the MMt.** Crucially, DES applied on PMC dramatically altered the patients' motor awareness, worsening their performance during the MMt. During the *online* MMt version of the task, four patients were stimulated in PMC and four patients in S1. In 88.9% of PMC trials (eight out of nine trials) affecting the HMt, the patients reported online that they were correctly executing the requested action (by saying OK), despite the complete arrest of their right-hand movement (white dots in Fig. 2b, c; Supplementary Movie 1). Conversely, DES delivered over S1 interrupted motor task execution without altering the patients' motor awareness. Indeed, in 100% of trials, the patients correctly reported the motor arrest (by saying STOP) (cyan dots in Fig. 2a; Supplementary Movie 2). Notably, during the *online* MMt, DES applied on PMC occasionally (four trials) induced a parallel disruption of phonoarticulation. Because this affects reliability of the motor monitoring, these trials were not included within the PMC percentage of significant trials reported above, therefore were excluded from statistical analysis. Interestingly, in the six excluded trials, patients reported normal motor execution when phonoarticulation returned, as in all other PMC trials.

The effect obtained on PMC during the *on-line* MMt was replicated in an additional four patients tested with the *delayed* MMt. All patients reported correct execution of the HMt (by answering YES) in the 100% of the trials (four out of four trials), despite complete movement arrest due to DES (red dots in Fig. 2c, d).

We compared PMC and S1 results by means of two sets of statistical analysis, either including all PMC trials, irrespective of the online/delayed version of the task or focusing only on the

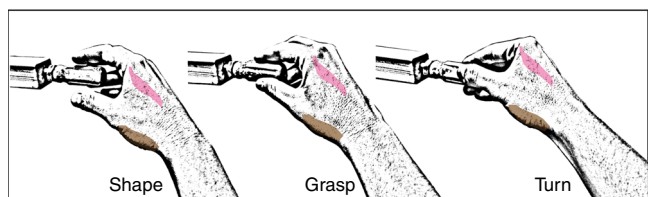

**Fig. 1 Hand-manipulation task.** Graphical reproduction of the motor task and its phases. The APB (abductor pollicis brevis) is shown in light brown and the FDI (first dorsal interosseous) in light pink.

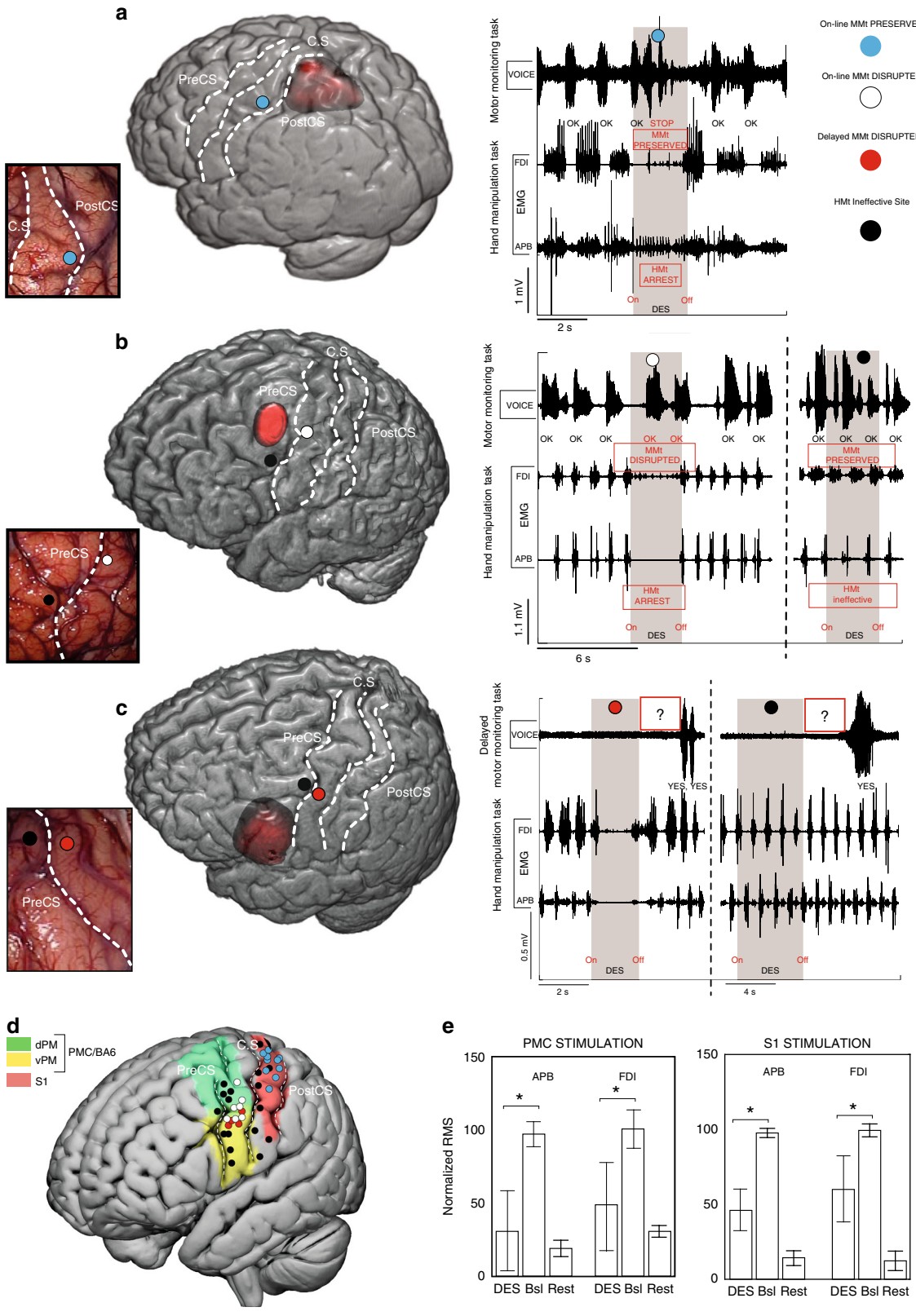

online trials. A significant association between the percentage of altered trials and the stimulated areas (PMC/S1) was found in both analyses (for both $p < 0.001$, Fisher's test), showing that altered motor monitoring specifically occurred only during PMC stimulation.

See Fig. 3 for detailed MMt results of each stimulation site in each patient.

**Disconnection analysis results.** Lesion-based disconnection analysis showed that the white matter tracts present in all five

**Fig. 2 Stimulation results. a** DES applied on the S1 hand area immediately anterior to the postcentral sulcus (PostCS) (cyan dot) interfered with hand-manipulation task (HMt), as showed by EMG activity of FDI and APB, but not with the online motor-monitoring task (MMt), as showed by the digitalized vocal signal, VOICE. **b** DES applied in PMC, immediately posterior to the precentral sulcus (PreCS), interfered with both HMt and with the online MMt (white dot). DES applied anteriorly to PreCS (black dot) did not interfere with either of the two tasks. **c** DES applied on PMC, immediately posterior to the PreCS (red dot), interfered with both the HMt and with the delayed MMt. DES applied anteriorly to the PreCS (black dot) did not interfere neither of the two tasks. In the delayed task, the question was asked immediately after DES-offset (red rectangle with question marker). **d** 3D template reporting all stimulation sites in the 12 patient sample. In all the brain renderings C.S indicates the position of central sulcus. **e** Statistical analysis comparing the muscle unit recruitment (RMS) during baseline (Bsl, HMt execution without DES) and during DES of S1 and PMC. The results show that generally DES in both PMC and S1 decrease RMS activity in both muscles with respect to the same muscles during baseline (whiskers indicate standard deviation, Kruskal–Wallis test, *$p < 0.05$).

patients with a probability level of 80% were nine, including the anterior thalamic radiation, arcuate fasciculus, corticofugal tracts, fronto-insular tracts, fronto-striatal tracts (FST), the inferior frontal occipital fasciculus, cortico-pontine tracts, and the superior longitudinal fasciculus II and III (SLF II and SLF III) (Fig. 4a). We used these tracts as templates to investigate the similarity between AHP patients and tracts disconnected virtually. PMC virtual disconnection analysis showed the involvement of five out of nine of the tracts considered (Fig. 4b). The common tracts were divided based on their level of probability: high (80%) and medium (40%). Tracts showing a high probability of disconnection included the arcuate fasciculus and both branches of the SLF II and III (Fig. 4c). Tracts of medium probability included the FrIns and the FST. Coherently they also showed a lower proportion of shared voxels with the virtual lesion volume. Regarding S1, virtual disconnection analysis showed that only the U-shaped fibers connecting S1 and M1 were significantly involved with a percentage over 80% (Fig. 4d). These fibers never highlighted in the AHP patients and PMC DES-related virtual lesion results. The SLF II and III were not totally absent, but they were involved in the virtual disconnection volume with a very low probability (below 20%). Taken together, this analysis suggested that long range temporo-parieto-premotor networks, mainly including the SLF II, III, and arcuate fasciculus are the main common tracts involved in both AHP patients and PMC DES-related virtual lesion results. A different network was involved when considering the control area, S1, mainly involving the short-range U-shaped fibers from postcentral to precentral gyrus related to primary sensorimotor representation of the hand.

## Discussion

In the present study, we directly tested the role of the PMC in conscious monitoring of voluntary actions. Our results show that DES was effective in interfering with motor execution when applied to both PMC and S1, but, crucially, it dramatically altered the patients' motor awareness only when applied to PMC.

The motor effect of both PMC and S1 stimulations resemble the negative motor responses, described by Luders et al.[19] as the cessation/arrest or decrease of the ongoing voluntary movement. It has been shown that negative motor responses can occur for both upper limb and speech-related movements[20] and that negative motor responses within BA6 can arrest both movements simultaneously[21]. Although the latter case was also observed in the present study (4 out of 17 trials), in most PMC stimulations (as well as in S1 stimulations), DES selectively blocked hand movements without affecting speech-related movements during the on-line motor monitoring task. Note also that, as shown by the results of the naming task administered during the DES procedure, when phonoarticulation was preserved, perseverations or other language-related impairments were never observed, thus making it unlikely that any differences in the online MMt observed between target and control areas might be ascribed to DES-induced language impairments.

Crucially, and in keeping with our hypothesis, only during PMC stimulations, in both online and delayed versions of the motor monitoring task, patients were unaware of the motor arrest and erroneously reported correct motor-task execution. In other words, they were not aware that the movement they had programmed was not executed. This behavior resembles the patho-logical condition of AHP patients (who claim to be able to move their paralyzed hand), thus showing an intraoperative virtual model of AHP. It is important to note that, in the clinical context, a dissociation between implicit and explicit aspects of motor awareness has been described in AHP patients[22–24]. For instance, some AHP patients approach ecological bimanual tasks (such as opening a bottle) adopting compensatory unimanual strategies (i.e. putting the bottle between the legs), thereby exhibiting some implicit knowledge of their motor deficit. The MMt, used in this study, based on the patients' verbal reports, was however designed to evaluate only explicit motor awareness and the present results have to be closely related to this aspect. Remarkably, even if AHP has traditionally been associated with right-brain damage[25], this disorder emerges also in left brain-damaged patients, at least when the assessment avoids language-related problems[26]. The present study, including only left brain patients due to clinical constraints (neurophysiological brain mapping is mandatory to preserve language-related functions), confirms that there is no absolute right-brain dominance for motor awareness. This might suggest that, at least in the motor context, the monitoring function is embedded in the same ipsilateral sensorimotor network controlling motor execution of the contralateral upper limb. To verify this hypothesis with DES, future experiments are needed to compare the effect of both right and left PMC stimulation on the monitoring of either contralateral or ipsilateral hand movements. Importantly, due to the clinical constraints of the awake brain surgery procedure, the present conclusions are based on a sim-plified experimental design (e.g. no catch trials were included) and on a relatively small number of stimulations for each site (13 for PMC in 8 patients; 10 for S1 in 4 patients); thus, future studies are needed to confirm these findings in a greater population. From a clinical point of view, the ability to preserve the areas involved in motor awareness is crucial. It is well known that AHP presents a significant risk for negative functional outcome in stroke rehabilitation[27]. Similarly, in brain tumor patients, the introduction of an MMt during intraoperative brain mapping is a promising tool to improve rehabilitation and functional recovery of patients.

It is well known that other brain areas, namely the cerebellum and the posterior parietal cortex (PPC), also play an important role in motor-monitoring during voluntary actions. These two regions may work in parallel to predict the sensory consequences of the movement and to make movement adjustments and cor-rections[28–30]. Interestingly, it has been suggested that on-line action control seems to be a specific function of PPC and that PMC is not actually involved in conscious motor monitoring. This conclusion is based on the observation that, during the

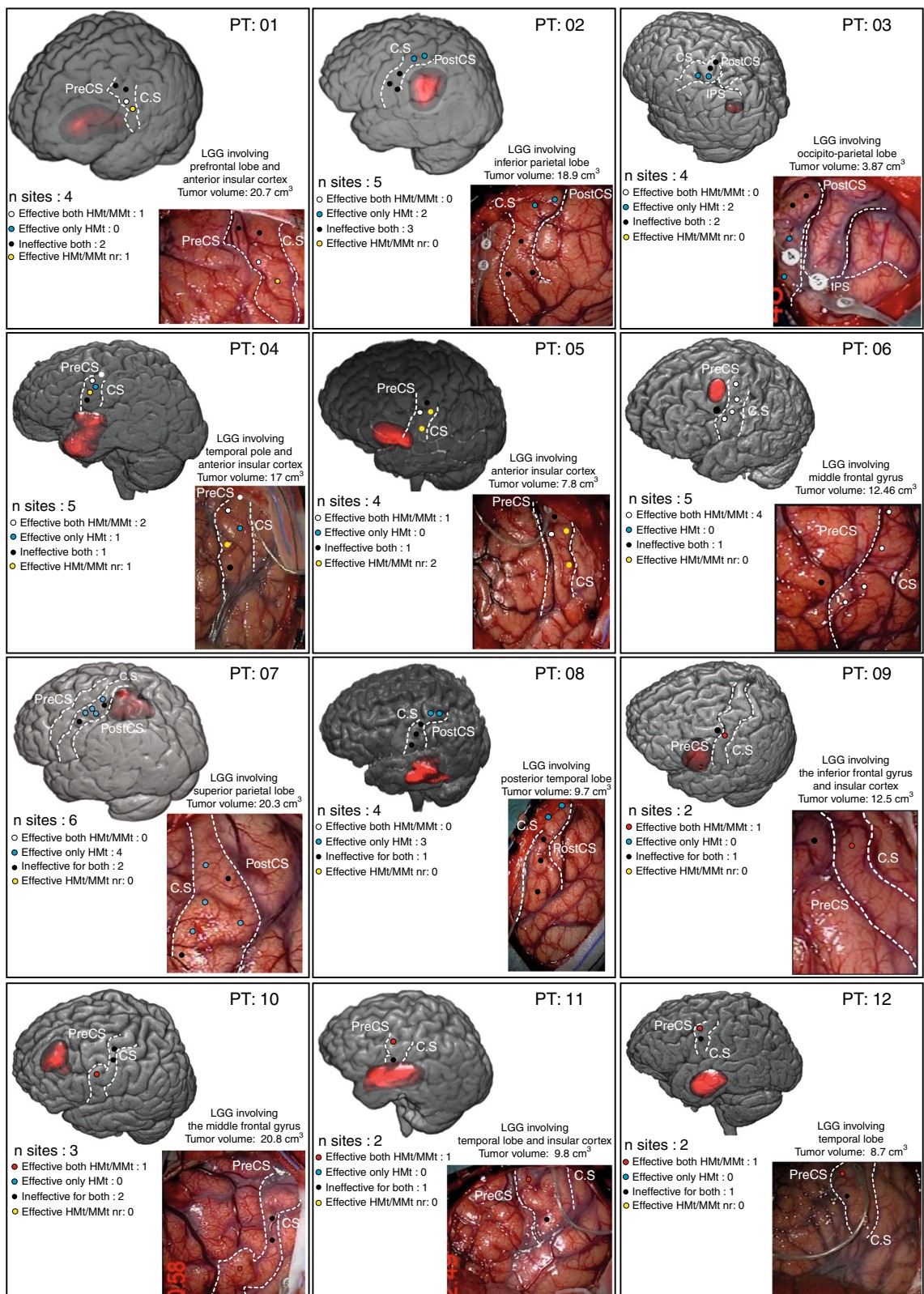

**Fig. 3 Population results.** Effective both HMt/MMt—red dot: DES disrupted both HMt and delayed MMt. Effective both HMt/MMt—white dot: DES disrupted both HMt and online MMt. Effective only HMt—blue dot: DES disrupted only the HMt. Effective HMt/MMt nr—yellow dot: DES disrupted the HMt and interfered with the phonoarticulation; therefore, the MMt was not reliable (nr). Ineffective for both—black dot: DES did not disrupt the tasks. Both the HMt and MMt were correctly executed.

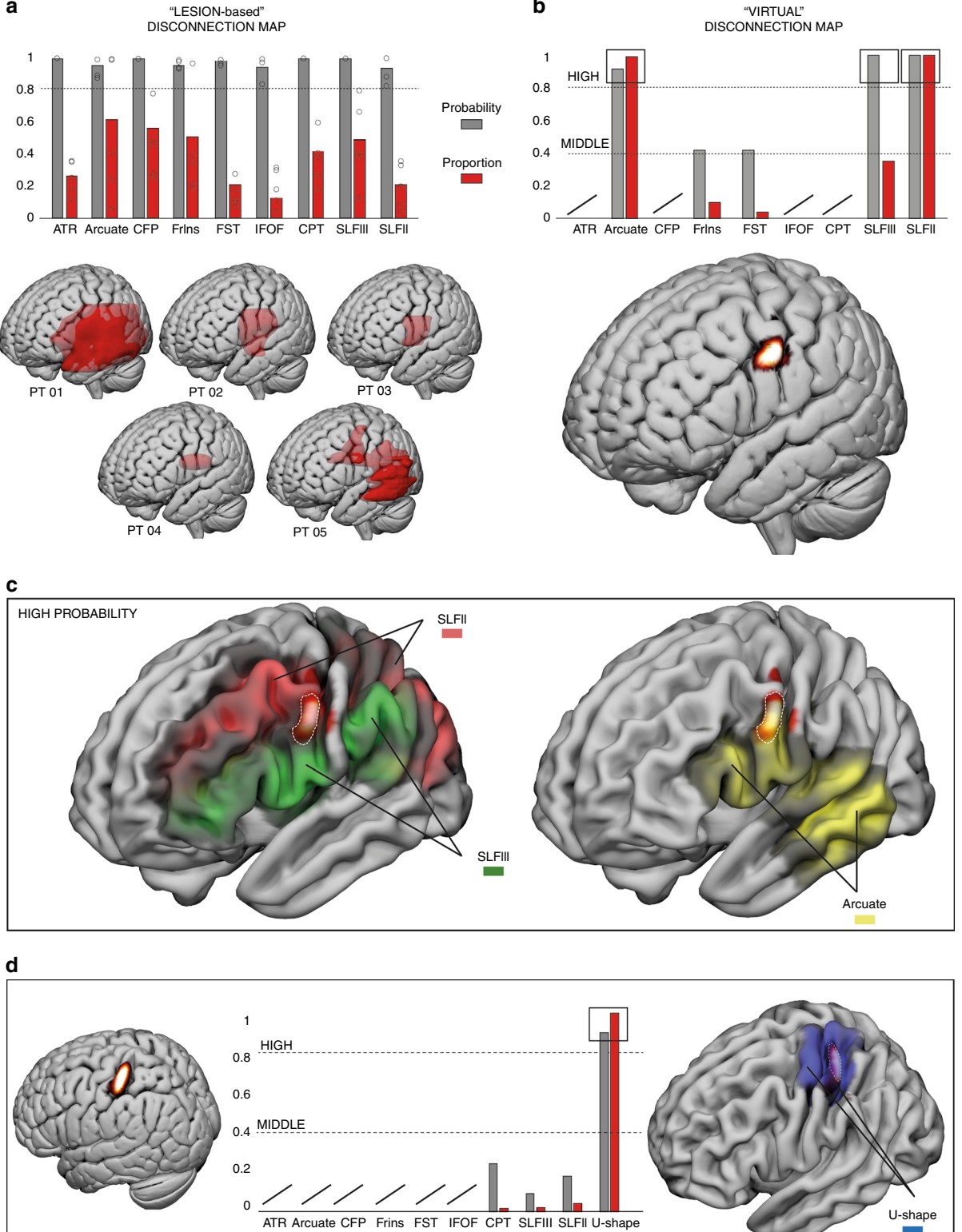

**Fig. 4 Disconnection results.** Results for the lesion-based (**a**) and virtual-lesion (**b**) disconnection analysis. In the bar graphs we showed the probability (gray) and the voxels proportion (red) of the tracts disconnected in all five AHP patients with a minimum probability of 80%. Below the bar graphs the lesion of the five AHP patients (**a**) (each circle-dot corresponds to patient probability for each tract) and the probability density estimation from which we extracted the virtual-lesion volume (**b**). Both are plotted on the ICBM152. **c** Anatomical relationship between the PMC virtual lesion and the SLF II–III and the arcuate plotted onto the smoothed ICBM152. **d** Anatomical relationship between the S1 virtual lesion and U-Shape fibers connecting postcentral and precentral gyrus.

patients' resting condition, DES on PPC induces a conscious motor intention, while DES on PMC evokes overt contralateral limb movements without a conscious motor experience[31]. However, alternative interpretations can account for this apparently contradictory finding. As a first explanation, the absence of motor awareness (in the presence of motor execution) could emerge from a lack of motor intentionality, since in Desmurget et al.'s study[32], movements were artificially generated by the PMC stimulation. As recently suggested, during involuntary movements, the comparator system in PMC is not activated, because without an intentional component, the motor system does not generate an efference copy to be compared with somatosensory feedback. To face this problem, the experimental protocol designed here was based on a voluntary motor task, thus ensuring motor intention which, in turn, triggers the cascade of events leading to the comparison between expected and actual motor feedback. Alternatively, it is possible that the lack of motor awareness induced by DES is, in fact, the expected outcome of stimulating a region involved in this type of function (i.e. motor monitoring). Coherently, in the present study, DES applied on PMC during a motor task simultaneously impaired action execution and action awareness. In a recent paper[21], Desmurget and colleagues also showed a selective inhibition of volitional hand movements after DES on PPC. They reported that, during these negative motor responses[19] in PPC, all patients were fully aware of their inability to move (as with our patients during S1 stimulations)[21]. This is supported by the few cases of motor monitoring during negative motor responses previously reported in the literature[33], even if the precise anatomical localization of negative motor areas is difficult to establish, thus making a direct comparison impossible with the present study. In our view, the recent finding of Desmurget and colleagues (i.e. preserved motor awareness during PPC negative motor responses) provides a further compelling control condition for the specificity of the effect we found with PMC stimulation (i.e. a parallel disruption of both the HMt and the motor monitoring task).

However, caution is needed in directly comparing previous studies with the present one, due to anatomical differences in stimulation sites over PMC[20,31]. In the present study, effective sites are located mainly in an area compatible with the potential homolog of the rostral sector of the non-human primate ventral PMC (F5)[34,35]. More precisely, our effective sites are located in the more dorsal sector of the ventral PMC, as shown in Figs. 2d and 4b, where the boundaries of the PMC subsectors are plotted according the template proposed by Mayka and colleagues[36]. Traditionally, different functional models of PMC contributions to sensorimotor behavior have been proposed according to its anatomical subsectors. For instance, the dorsal PMC appears to be mainly involved in guiding advanced motor planning according to the desired end state of an action[37], whereas ventral PMC appears to be involved in sensorimotor integration for grasping and controlling the kinematic synergies for an appropriate hand-object interaction[37,38]. Coherently, DES on the ventral PMC disrupted the hand-manipulation movements required by the HMt. However, this effect (i.e. HMt disruption) was not specific for the PMC (target area), but a similar effect in terms of muscle unit recruitment and task disruption was obtained also by DES on S1 (control area) (see Fig. 2a, e). These comparable motor effects, despite the anatomo-functional organization of the two areas would suggest different roles in the voluntary hand movements, might be due to the direct connections of both ventral PMC and S1 with the hand-finger sector of M1[39]. In the control condition, the disconnection analysis indeed clearly shows a highly involvement of the U-shaped fibers connecting S1 and M1 (see Fig. 4d). Despite the lack of conclusive evidence with a direct approach of the anatomo-functional

connections between ventral PMC and M1, some studies using an indirect approach, such as conditioning TMS studies, suggested a significant functional direct interaction between ventral PMC and M1 both at rest and during visually guided grasping[40,41] reasonably occurring via direct connections. Moreover, in non-human primate it is well known that ventral PMC is strongly connected with the M1 hand representation[42] and significantly affects its motor output[43].

Crucial for the purpose of the present study, the disconnection analyses applied on both virtual and actual lesions producing motor awareness deficits showed the significant presence of three common white matter tracts including the arcuate fasciculus, the SLF II and the SLF III (see Fig. 4a). This structural temporo-fronto-parietal connectivity pattern is highly compatible with the role of ventral PMC as a comparator system[2,7–9], proposed by lesion studies in AHP patients[2,7–9] and by neuroimaging studies in healthy subjects[32,44–47]. For instance, ventral PMC has been recently described as the neural correlate of motor-monitoring during mechanical limb immobilization. The increased ventral PMC activity during impossible movements has been related to conscious detection of the mismatch between movement planning and (no) movement execution[44]. According to this comparator model the hypothesis is that once the motor command is ready to be issued, an efference copy of the movement is generated in motor planning areas (e.g., SMA and dorsal PMC) and, on the basis of this signal, a forward model predicts the sensory consequences of the action. When the movement is implemented, the afferent inputs coming from muscle contractions reach S1, which, in turn, send sensory feedback to PPC, where sensory information converges. Finally, the sensory information of the movement is sent, through fronto-parietal connections, from the PPC, to the ventral PMC, which should compare the actual sensory input with the sensory predictions generated by the forward model[32,44]. This comparison may eventually lead to a veridical motor awareness (in our context, it leads the patient to recognize, as during S1 stimulation, that no movement has been performed and to say STOP). However, when the comparator system is altered by real or virtual lesions in PMC, a non-veridical motor awareness is generated (this leads the patient to believe that the hand is moving, despite the motor arrest, and to say OK or YES, depending on the online/delayed version of the MMt).

To sum up, our results indicate that, during voluntary hand movements, DES on both PMC and S1 interrupted movement execution, while only DES applied on PMC dramatically altered the patients' motor awareness, making them unconscious of the motor arrest. Taken together, these findings promote the role of PMC as a shared neural substrate for both motor execution and motor awareness of voluntary actions, disclosing a crucial hub in the anatomo-functional network of human motor awareness.

## Methods

**Patient selection**. In this study, we included 12 patients affected by low grade gliomas (LGGs), candidates for surgery requiring the exposure of the frontal or parietal lobe. Each patient underwent an extensive and multidisciplinary pre-operative study (see below). All patients gave written informed consent to the surgical and mapping procedure, and to data analysis which followed the principles outlined in the World Medical Association Declaration of Helsinki: Research involving human subjects. The study was approved by the Ethics Committee of the Humanitas Research Hospital (protocol IRB n.001299) and performed with strict adherence to the routine procedure normally employed for surgical tumor removal. Accordingly, all data were recorded using electrophysiological monitoring and stimulating protocols (see below) adopted for routine clinical mapping.

**Pre-surgical routine**. Preoperatively, all patients were assessed for handedness, underwent a neurological examination and a neuropsychological evaluation. Patients without language and apraxia symptoms (assessed by neuropsychological evaluation) and without motor and somatosensory deficits (assessed by neurological examination) were considered candidates for the study. All patients enrolled

## Table 1 Socio-demographic and clinical data.

| ID | Age | Gender | Education | Hand | Location | Laterality |
|----|-----|--------|-----------|------|----------|------------|
| 1  | 55  | F | 17 | R | F | L |
| 2  | 30  | M | 17 | L | P | L |
| 3  | 27  | M | 13 | R | P | L |
| 4  | 58  | M | 17 | R | F | L |
| 5  | 32  | F | 13 | R | F | L |
| 6  | 52  | F | 17 | R | P | L |
| 7  | 54  | M | 17 | R | F | L |
| 8  | 44  | F | 13 | R | T | L |
| 9  | 34  | M | 13 | R | F | L |
| 10 | 46  | M | 13 | R | F | L |
| 11 | 48  | M | 13 | R | F | L |
| 12 | 58  | M | 13 | R | F | L |

ID, identification number; gender: F female, M male; hand, handedness; location tumor location: F, frontal, P, parietal; laterality: R, right, L, left.

## Table 2 Preoperative cognitive performance.

| ID | Naming test | Token test | Ideomotor apraxia test |
|----|-------------|------------|------------------------|
| 1  | 47.00 (ES = 3) | 34.25 (ES = 4) | 71 (ES = 4) |
| 2  | 44.91 (ES = 2) | 36.00 (ES = 4) | 72 (ES = 4) |
| 3  | 48.00 (ES = 4) | 36.00 (ES = 4) | 72 (ES = 4) |
| 4  | 46.69 (ES = 3) | 36.00 (ES = 4) | 72 (ES = 4) |
| 5  | 45.97 (ES = 2) | 31.5 (ES = 3) | 72 (ES = 4) |
| 6  | 45.57 (ES = 3) | 36.00 (ES = 4) | 72 (ES = 4) |
| 7  | 48.00 (ES = 4) | 36.00 (ES = 4) | 72 (ES = 4) |
| 8  | 48.00 (ES = 4) | 36.00 (ES = 4) | 72 (ES = 4) |
| 9  | 46.69 (ES = 4) | 36.00 (ES = 4) | 72 (ES = 4) |
| 10 | 45.97 (ES = 4) | 31.5 (ES = 4) | 72 (ES = 4) |
| 11 | 45.57 (ES = 4) | 36.00 (ES = 4) | 72 (ES = 4) |
| 12 | 48.00 (ES = 4) | 36.00 (ES = 4) | 72 (ES = 4) |

The table report the age/education-adjusted score and the equivalent score (ES) for each subject and for each test. ES = 0: deficit, ES = 2, 3, 4: normal performance. ID, identification number.

in the study scored within the normal range for all the parameters assessed in both evaluations (see Tables 1 and 2). Preoperative magnetic resonance imaging (MRI) was performed using a Philips Intera 3T scanner (Best). The neuroradiological examination included basic morphological T1, T2, FLAIR, DWI, and post-contrast T1 images[13]. Tumor volume, including possible edema, was computed on the volumetric Fluid-Attenuated-Inversion-Recovery (FLAIR) MRI scans, as routinely applied for LGGs. Twelve right-handed patients were enrolled in the study, all candidates for surgical removal of a tumor located in the left hemisphere (eight required exposure of the frontal lobe while four required exposure of the parietal lobe). Notably, only patients with LGGs not infiltrating the target/control area were included in the study, for which borders of the tumors were located at a minimum distance of 1 cm from the target/control area, to minimize possible misleading effects due to tumor infiltration. During preoperative assessment, all patients were trained to perform the experimental tasks (see below) routinely executed during intraoperative investigation without introducing variations in clinical procedures.

**HMt and MMt**. The HMt consisted in the rhythmical manipulation of a small cylindrical handle (∅ 2 and length 6 cm) inserted inside a fixed rectangular base (3 × 3 cm and 9 cm of length) by means of a worm-screw. The rectangular base was kept stable close to the patient's hand along the armrest of the operating table, while the patient grasped, held, and rhythmically rotated and released the cylindrical handle with the thumb and the index finger, resembling a precision grip. Proximity between the hand and fingers and the cylindrical handle allowed the patient to perform the movement using only the digits, avoiding a reaching movement (see Fig. 1). We asked the patient to perform the task following an internally generated rhythm in the absence of visual information about hand movement and the cylindrical handle (i.e., haptically driven). Each patient was trained one day before surgery by the neuropsychologists: the patient was asked to lie in a bed, in a position resembling the intraoperative one and to execute the HMt.

In order to investigate the patient's ability to monitor his/her motor performance, the HMt was coupled with a verbal MMt.

We adopted two versions of the MMt: the *online* MMt and the *delayed* MMt. In the *online* MMt, patients (n = 8, four during PMC stimulation and four during S1 stimulation) were asked to verbally monitor the task overtly, in real time, by saying OK for each grasp-hold-turn phase executed without any difficulty, and by saying STOP when they experienced difficulties in task execution. In order to allow the patients to be familiarized with the coupled HMt and the on-line MMt task, the neuropsychologist mechanically blocked the cylindrical handle while patients were performing the HMt in random trials during the preoperative training. In this way, they were trained to monitor a disturbance in task execution by saying STOP. Importantly, since the intraoperative setting did not allow the patients to see their hand, preoperative training was performed blindfolded. At the end of the preoperative session all the patients were able to perform and monitor the HMt accurately.

In the *delayed* MMt, patients (n = 4) were asked to answer immediately after DES in PMC to a specific question: Did you correctly execute the motor task? The patient had to answer YES in the case of correct performance and NO in the opposite case. For the on-line MMt, the manipulandum was mechanically blocked in random trials during the preoperative training, allowing patients to familiarize with the task by answering NO whenever they noticed some difficulty in executing the task. As a control, the same question was asked occasionally during the intraoperative session, even during stimulation in areas adjacent to the PMC that did not evoke interference with the movement (see Fig. 2d).

The manipulandum movements were video recorded in order to measure the patients' performance during the task and to classify trials in which hand movements on the manipulandum were either correct or suppressed.

**Selection criteria for brain areas**. Target and control areas were selected for showing, during DES, comparable effects on motor execution and muscle recruitment (i.e. motor arrest during voluntary movements) and, possibly, a different effect on motor awareness (expected to be altered in the target area and preserved in the control area).

The ventral sector of the PMC was selected as a target area according to the following:

(1) A positive effect in altering motor execution was expected based on previous literature employing DES during the same HMt[12,14,15].
(2) A positive effect in altering the motor awareness was expected based on anatomical and functional evidence in AHP patients[2,7–9,48,49] and in healthy subjects[44,45]. It is important to note that lesion studies in AHP patients also highlight the role of other brain areas putatively relevant for motor unawareness, such as the temporo-parietal junction[48] and the insula[48,50]. Here, we a priori focused on the role of PMC.

The hand and finger sector of S1 was selected as a control area due to the following:

(1) Based on the literature[12,51] and on clinical experience, DES applied to the S1 region, immediately posterior to the hand-knob, evokes positive effects during hand-manipulation movements, altering motor execution similarly to the positive effect obtained in ventral PMC in terms of motor unit recruitment. The similar motor effect obtained from both areas was crucial in order to avoid possible confounding effects due to involuntary muscle contraction as often occurs when DES is applied in the hand-knob, hosting M1 hand representation and the main corticofugal pathway.
(2) A negative effect in inducing a motor awareness alteration was expected based on previous literature in both normal subjects[18,52] and AHP patients[7], suggesting that motor awareness can be independent from the somatosensory feedback computed within S1.

**Surgical procedure and routine intraoperative protocol**. The intraoperative protocol included asleep–awake–asleep anesthesia and functional brain mapping by means of electrophysiological and neuropsychological investigation[12,53,54]. Total intravenous anesthesia with propofol and remifentanil was used and no muscle relaxants were employed during surgery to allow mapping of motor responses. A craniotomy tailored to expose the cortex corresponding to the tumor area and a limited amount of surrounding tissue was performed. In every patient the surgical resection was performed with the aid of the intraoperative neurophysiological brain mapping and monitoring technique (see below). Cortical mapping using direct electrical stimulation (DES) was performed to define the safe cortical point of entry, while subcortical brain mapping was performed along with tumor resection, following the principle of locating functional boundaries which represented in all cases the limit of tumor resection[13]. Clinical constraints (different craniotomies in different procedures) prevented the simultaneous exposition, and testing, of the ventro-lateral PMC and the hand/fingers sector of S1 in the same patient. See Anatomo-functional reconstruction section for individual anatomical reconstruction of PMC and S1 sites.

**Neurophysiological monitoring**. During surgery, cortical activity was monitored by electroencephalography and electrocorticography (EEG, ECoG, Comet, Grass); ECoG from a cortical region adjacent the area to be stimulated was recorded by

subdural strip electrodes (4–8 contacts, monopolar array referred to a mid-frontal electrode) throughout the whole procedure, to monitor the basal electrical activity of the brain and to detect after-discharges or electrical seizures during the resection. EEG was recorded with electrodes placed over the scalp in a standard array. EEG and ECoG signals were bandpass filtered (1–100 Hz), displayed with high sensitivity (50–150 and 300–500 µV/cm, respectively), and recorded. The integrity of descending motor pathways was monitored throughout the procedure using a train-of-five stimuli delivered to the primary motor (M1) cortex to elicit motor-evoked potentials (MEPs) recorded in face, hand, and leg contralateral muscles, but was suspended during cortical and subcortical mapping to avoid interference. To this aim, a four-contact subdural strip electrode was placed over the precentral gyrus; each contact was tested, with a vertex reference, by stimulation with trains of 3–5 constant current anodal stimuli (pulse duration: 0.5–0.8 ms; interstimulus interval (ISI): 2–4 ms) at a repetition rate of 1 Hz. Electromyographic (EMG) responses to stimulation of the motor areas, as well as voluntary motor activity, were recorded throughout the procedure by pairs of subdermal hook needle electrodes (Technomed) inserted into 20 muscles (face, upper and lower limb) contralateral to the hemisphere to be stimulated, plus four ipsilateral muscles, all connected to a multichannel EMG recording system (2000 Hz sample frequency, ISIS-IOM, InomedGmbH)[13]. Free-running EMG was used to record responses to stimulation and to distinguish between electrical and clinical seizures. Simultaneously to the EMG signals from the different muscles, the vocal emission of the patients was also recorded by means of a microphone. In compliance with the ethical regulations, the patient's audio file of the voice during MMt could not be recorded during the intraoperative procedure: the voice recorded from the microphone was amplified by the EMG system in order to exclude the real audio sound and to represent the patient' voice as a change of voltage in the time domain (digitalized vocal signal). The digitalized vocal signal was used to match the intraoperative assessment of the MMt offline.

**Neurophysiological brain mapping**. Routinely, two stimulation techniques were used during neurophysiological brain mapping: Low Frequency (LF-DES) and High Frequency (HF-DES) protocol, according to the frequency of stimulation pulses delivered[13]. HF-DES was delivered through a monopolar probe (straight tip, 1.5 mm diameter (Inomed), with reference/ground on the skull overlying the central sulcus). HF-DES was delivered in trains of 1 to 5 constant anodal current pulses (pulse duration 0.5 ms; interstimulus interval, ISI: 3–4 ms) at a variable intensity (1–40 mA) depending on the patient's cortical excitability. The choice of technique used by the surgeon depends on the clinical setting and on the structure to be explored[12,13,35,55]. Note that, during the present experimental tasks (HMt coupled with MMt), only the LF-DES paradigm, routinely used during mapping of language and cognitive functions[13], was used. LF-DES consisted of trains of biphasic square wave pulses lasting for 2–5 s (0.5 ms each phase) at 60 Hz (ISI 16.6 ms) delivered by a constant current stimulator (OSIRIS-NeuroStimulator) integrated into the ISIS-System through a bipolar probe (2 ball tips, 2 mm diameter, separation 5 mm). Stimulation with a bipolar probe is rather focal[13] such that the effect obtained, i.e. an interference with the running task, is supposed to be due to the transient inactivation of a small area of tissue surrounding the probe. As a routine procedure, when mapping is performed with LF-DES, independent of tumor location (parietal, frontal, or temporal), language tasks (counting and naming) were tested first, followed by the HMt coupled with MMt. Intensity of LF-DES (the so-called working current, WC) was set to the threshold intensity needed to disrupt/arrest counting.

**Intraoperative procedure**. During execution of the HMt, patients' motor performance and verbal monitoring were recorded online and compared offline by two trained neuropsychologists. At the beginning of the HMt intraoperative session, 10 s of movement baseline (without stimulation) coupled with correct MMt was mandatory to reach stable task execution (adaptation phase). Depending on clinical needs, in some cases the adaptation phase lasted more than 10 s. In order to verify, intraoperatively, the correct execution of the online MMt by the patient, just before the beginning of the DES mapping session a mechanical block of the cylindrical handle was evoked, suddenly, by the neuropsychologist. All patients included in the study correctly performed this trial by reporting the expected STOP when the block occurred. Similarly, in the retrospective motor monitoring a mechanical block of the cylindrical handle was evoked, in the same way, and immediately after the question "Did you correctly execute the task?" was asked. All patients included in the study correctly responded NO to this trial. Note that, to exclude that verbal responses during the MMt could be biased by speech impairment, all sites included in the analysis were stimulated, according to the procedure, during counting and naming tasks. In agreement with what was observed for the MMt, speech arrest occurred in 22.2% of PMC stimulations (6 out of 27 sites), while it never occurred during S1 stimulations (0 out of 20 sites). Importantly, in the PMC trials where phonoarticulation was preserved (88%), perseverations, repetitive verbalizations, or others speech impairments were never observed.

When a satisfactory baseline was reached in both behavioral outcome and motor monitoring performance, the surgeon started the mapping procedure by delivering LF-DES over the cortical areas surrounding the tumor with a stimulation interval among the different sites of 2–3 s, in order to avoid a possible dragging

effect. Simultaneously, two trained neuropsychologists reported the observable hand behavioral outcome during stimulation to the surgeon and wrote down the verbal responses of the patients during the MMt.

**Anatomo-functional reconstruction**. For each patient, the reconstruction of the exact position of the stimulated sites was computed. During intraoperative mapping, the entire exposed craniotomy was video recorded and the MRI coordinates of the sites were acquired by the neuronavigation system. To determine the exact position of the sites on the three-dimensional (3D) MRI cortical surface of each patient the following procedure was adopted. The post-contrast T1-weighted sequence of each patient (the same loaded into the neuronavigation system during surgery) was used to perform the cortical surface extraction and surface volume registration computed with the FreeSurfer Software. Subsequently the results were loaded in Brainstorm (MatLab Tool Box), which is a software documented and freely available for download online under the GNU general public license (http://neuroimage.usc.edu/brainstorm). With the aid of Brainstorm, the exact position of the site coordinates was marked as a scout on the patient's 3D MRI and the labeled scout were co-registered to the MNI space system by means of the unified segmentation implemented in SPM12. The coordinates of each site were then entered into the ICBM152 template to create a 3D reconstruction of the left (stimulated) hemisphere.

**Statistical analysis**. A Fisher's exact test was used to compare the effectiveness of DES in interfering with the MMt in the target and control areas (PMC vs S1). As a dependent variable we used the patients' dichotomous response for the MMt (OK/STOP for online MMt and YES/NO for the delayed MMt) in two sets of analyses, either including all PMC trials, irrespective of the online/delayed version of the task or focusing only on the online trials. A significant association between the percentage of altered MMt trials and the stimulated areas (PMC vs S1) was expected to confirm our predictions. This target/control design rules out important confounding variables, since S1 stimulations induced a comparable motor arrest as compared to PMC stimulations, producing the same noise therefore identical in terms of patients' expectation. EMG analysis was also performed in order to assess the effect of DES on task performance considering all the effective sites in S1 and PMC. Specifically, the amount of muscle units recruited (measured as root mean square, RMS) recorded during HMt was compared between two conditions: (1) RMS during DES in PMC (17 trials) and S1 (10 trials) vs (2) RMS during Baseline (corresponding to 4 s of HMt execution before effective stimulation) by means of a Kruskal–Wallis test followed by multiple comparison of mean ranks. To do that the RMS signal was normalized within muscle and patient by dividing the RMS activity during DES in PMC/S1 and a baseline of RMS activity recorded during HMt execution before starting the mapping procedure.

**Disconnection analysis**. In order to investigate the neural network belonging to the target (PMC) and control (S1) areas we additionally applied an indirect structural approach[56] to our data in which is possible to identify key white matter tracts affected by DES (virtual structural disconnection) or by a specific lesion (lesion-based structural disconnection). To do that we performed a probability density estimation of the stimulated sites in order to translate the coordinates of the stimulated positive sites to a volume[57] corresponding respectively to the most effective PMC region tested (PMC virtual lesion volume) and to the most effective S1 region tested (S1 virtual lesion volume). In order to highlight the most probable white matter tracts and its relative proportion belonging to the positive tested regions, the PMC virtual lesion volume was extended in order to include the surrounding white matter until the fundus of the precentral sulcus, while S1 virtual lesion volume was extended until the fundus of the postcentral sulcus. Similarly, we also applied lesion-based disconnection analysis in five AHP patients (5 AHP patients[4,9]) in order to study the main white matter tracts involved in all the patients included with a high probability level (from 80%). Patients' neuroimaging data were acquired by means of computerized tomography (CT) or MRI-FLAIR and lesions were segmented with MRIcron software. Normalization in the MNI space was computed using Clinical Toolbox in SPM12. The same tracts highlighted by the lesion-based analysis in AHP patients were specifically investigated in the virtual disconnection analysis obtained by DES in PMC and S1. Since, in the AHP patients, the lesions were located in the right hemisphere we flipped the hemisphere orientation in order to be comparable with the results of the DES study obtained in the left hemisphere. As a control we performed the structural disconnection also for the patient's lesion in their native space.

**Reporting summary**. Further information on research design is available in the Nature Research Reporting Summary linked to this article.

## Data availability
The data that support the findings of this study are available on request from the corresponding author. A reporting summary for this article is available as a Supplementary Information file.

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

## Acknowledgements

The authors are grateful to all the patients involved in the study. We are also in debt to Giacomo Rizzolatti, Paola Borroni, and Marco Neppi-Modona for their precious comments on the first draft of the manuscript and to Henrietta Howells for her helpful input. This work was supported by the MIUR-SIR 2014 grant (RBSI146V1D) and by the San Paolo Foundation 2016 grant (CSTO165140) to F.G.

## Author contributions

F.G., A.B. and G.C. conceived the study; F.G. received the funding to support the study; F.G., G.C., L.F. and G.P. designed the experimental paradigm; L.B. selected patients, directed, and executed the surgical procedure and the intraoperative brain mapping and contributed to the 3D reconstruction.; L.F.,G.P., L.B. and A.L. acquired data, analyzed data, and produced figures; all the author discussed and interpreted the results; F.G., L.F., G.P. and G.C. wrote the first draft of the paper; all the authors read the paper and contributed to the final draft.

## Competing interests

The authors declare no competing interests.
