## [Peer Review File · Nature Communications]

Editorial Note: Parts of this Peer Review File have been redacted as indicated as we could not obtain permission to publish the reports of Reviewer 1.

Reviewers' Comments:

Reviewer #1:
Remarks to the Author:

[Redacted]

Reviewer #2:
Remarks to the Author:

The present article by Fornia et al. uses intra-operative cortical stimulation to determine the functional role of the premotor cortex for motor-monitoring and awareness of movements in n=12 brain tumor patients. They applied an inhibitory stimulation to the PMC and were able to demonstrate that this stimulation inhibited the motor performance and the awareness that the movement was not performed correctly. As control condition they applied the stimulation to S1, where they could also impair the movement execution, here in contrast to the PMC the subjects were aware that they did not perform the movement. The present study adds by a direct virtual lesion approach significantly to the understanding of motor-monitoring /awareness and provides causal support for the concept that the PMC plays a crucial role in this functions, as suggested previously. The study was well performed in n=12 patients, the methods are sound. It would be helpful to see the data of each individual patient (was the behavioral effect apparent in all patients? All in the same location?) and a more detailed description of the statistical approach and results. Overall it is a well-performed study using intraoperative stimulation with behavioral evaluation to demonstrate/corroborate in a causal fashion the functional role of the PMC for motor-monitoring and motor-awareness. A more detailed discussion of the area they found to be relevant within BA6 should be added in the view of the different functional parts of the vPM/lateral BA6.

Suggestion: invitation for revision

Reviewer #3:
Remarks to the Author:

We read with great interest the manuscript by Garbarini et al who examined a fascinating topic through a creative and original experiment. In particular, we find relevant the attempt to reproduce "in vivo" through direct electrical stimulation (DES) a model for studying motor awareness. The authors show a dissociation in the motor awareness of ongoing grasping hand movements between premotor (PM) and somatosensory (S1) cortex stimulation. While DES of both regions impaired movement, only DES of PM impaired awareness of movement. This finding causally demonstrates the involvement of premotor cortex in motor awareness consistent with impairment of awareness (anosognosia for hemiplegia) after strokes of premotor regions, and models in which awareness depends on the generation of premotor plans rather than sensory feedback. The paper is unique and novel in the direct manipulation of motor awareness via physiological methods. It is also relevant to the current debate about the hemispheric lateralization of anosognosia for hemiplegia as a right hemispheric syndrome.

Nonetheless, we have some concerns:

- 1) The study was conducted on patients with left brain glioma requiring awake surgery. One possible concern is that the location of PMC was displaced or reorganized by the neighboring tumors. Gliomas are known to lead to a local reorganization of function. It would be helpful to see the tumor location, also in term of edema, vis-à-vis the location of 'active' sites. That could be done by segmenting the tumors plus edema and plotting the lesion individually with the active DES sites. Is there any evidence of displacement? Are we confident that PMC is indeed the correct localization? Finally, please report tumor's grade, and size in relation to neurological and neuropsychological profiles for each patient.
- 2) The approach is strongly guided by a-priori hypothesis about the role of PMC and S1. Please provide a justification for the selection of S1 (another reasonable choice would have been M1), as compared to other regions implicated in anosognosia (e.g. TPJ or insula).
- 3) PMC is a big place. The sites seem to be located in ventral PMC. It would be helpful to see the sites on more precise maps of PM (e.g. the Julich atlas). This is important for the interpretation. Specifically, what is the relationship of these premotor and S1 sites vs. M1 sites that produce also motor deficits? I am wondering if the PM sites were 'hand' related or perhaps more generally involved in 'action' planning of hand gestures, or some higher planning function? This is relevant since some sites appeared to be interfering with the verbal output indicating a mouth representation.
- 4) A related approach to solve these issues would be to try to use the location of seeds, as compared to control seeds, to map functionally or structurally the networks involved. This could be done with indirect structural functional disconnection approaches as proposed by Thiebaut de Schotten (Foulon et al.) and Fox and colleagues (Boes et al.)
- 5) I am skeptical that the 'negative' sites did not produce any EMG or motor effect. It would be important to see a quantitative analysis of this point. Can you report on movie or EMG data on this issue?
- 6) Do the authors have an hypothesis on why subjects reported correctly about 15% of the trials during PMC stimulation? Was the EMG different in those trials? Was there a threshold effect?
- 7) A related issue is explicit vs. implicit motor awareness. Please comment whether the authors believe this distinction is purely quantitative (more or less 'intention' become verbally aware) or reflect distinct processes.
- 8) Anosognosia for hemiplegia is more common after right hemisphere lesions. Do they expect to see the same effect for right PM DES but in the left arm? Shall we then think of the neuropsychological syndrome as reflecting an impairment of a higher level monitoring system that is distinct from the contralateral motor/premotor system?

Maurizio Corbetta

RESPONSE

We thank the Editor and the Reviewers for their positive comments and for their constructive criticism. We incorporated all their suggestions (including new analysis, new figures and new tables) in the revised version that has been substantially improved. Note in red color the modified part of the main text.

Reviewer #1 (Remarks to the Author):

[Redacted]

RESPONSE

[Redacted]

Reviewer #2 (Remarks to the Author):

The present article by Forna et al. uses intra-operative cortical stimulation to determine the functional role of the premotor cortex for motor-monitoring and awareness of movements in n=12 brain tumor patients. They applied an inhibitory stimulation to the PMC and were able to demonstrate that this stimulation inhibited the motor performance and the awareness that the movement was not performed correctly. As control condition they applied the stimulation to S1, where they could also impair the movement execution, here in contrast to the PMC the subjects were aware that they did not perform the movement. The present study adds by a direct virtual lesion approach significantly to the understanding of motor-monitoring /awareness and provides causal support for the concept that the PMC plays a crucial role in this functions, as suggested previously. The study was well performed in n=12 patients, the methods are sound.

It would be helpful to see the data of each individual patient (was the behavioral effect apparent in all patients? All in the same location?) and a more detailed description of the statistical approach and results.

RESPONSE

We thank the Reviewer#2 for this suggestion and, as also suggested by Reviewer#1, we added in the revised version a Table including details about data of each individual patient. See Table 2. The table clearly show that, when lateral BA6 was stimulated, approximatively at the same location, behavioral outcome was similar for all patients.

We also added details on the statistical approach to make clearer our results. See Methods, p. 13, lines 20-25: p. 14, lines 1-3.

Overall it is a well-performed study using intraoperative stimulation with behavioral evaluation to demonstrate/corroborate in a causal fashion the functional role of the PMC for motor-monitoring and motor-awareness.

RESPONSE

We thank the Reviewer#2 for this positive comment.

A more detailed discussion of the area they found to be relevant within BA6 should be added in the view of the different functional parts of the vPM/lateral BA6.

RESPONSE

We agree with the Reviewer#2 that lateral BA6 has different functional parts and, as suggested, we added a more detailed discussion about the location of our stimulation sites in PMC, also in agreement with the results of the additional disconnection analysis suggested by the Reviewer#3 (p. 20 lines 7-33:p. 21 lines 1-16). Furthermore, as also suggested by Reviewer#3 (see point 3), in the revised version of Figure 2, we plotted the boundaries of the PMC subparts proposed by Mayka et al (2006) human motor area template (HMAT).

Suggestion: invitation for revision

Reviewer #3 (Remarks to the Author):

We read with great interest the manuscript by Garbarini et al who examined a fascinating topic through a creative and original experiment. In particular, we find relevant the attempt to reproduce “in vivo” through direct electrical stimulation (DES) a model for studying motor awareness. The authors show a dissociation in the motor awareness of ongoing grasping hand movements between premotor (PM) and somatosensory (S1) cortex stimulation. While DES of both regions impaired movement, only DES of PM impaired awareness of movement. This finding causally demonstrates the involvement of premotor cortex in motor awareness consistent with impairment of awareness (anosognosia for hemiplegia) after strokes of premotor regions, and models in which awareness depends on the generation of premotor plans rather than sensory feedback.

The paper is unique and novel in the direct manipulation of motor awareness via physiological methods. It is also relevant to the current debate about the hemispheric lateralization of anosognosia for hemiplegia as a right hemispheric syndrome.

RESPONSE

We thank very much the Reviewer#3 for this positive comment, we are very pleased to know that he found our work “unique and novel”.

Nonetheless, we have some concerns:

1) The study was conducted on patients with left brain glioma requiring awake surgery. One possible concern is that the location of PMC was displaced or reorganized by the neighboring tumors. Gliomas are known to lead to a local reorganization of function. It would be helpful to see the tumor location, also in term of edema, vis-à-vis the location of ‘active’ sites. That could be done by segmenting the tumors plus edema and plotting the lesion individually with the active DES sites. Is there any evidence of displacement? Are we confident that PMC is indeed the correct localization?

RESPONSE

We understand the Reviewer#3’s concern about a possible displacement of the PMC function. This eventuality has to be seriously considered when brain-tumor patients are used to make inferences about brain functions. We acknowledge that our patients’ description in Methods was not clear and that important details about tumor’s grade and size were missing, as also noticed by the Reviewer#3 in the point below. In the revised version, we clearly stated that all patients here had a low grade glioma LGG (see p. 6 line 4) and that in the tumor segmentation we included all the FLAIR alteration, possibly including edema (see p. 6 lines 20-21), that is, however, less relevant in LGG than in HGG patients. Furthermore, we added that the tumors in all patients were located at a minimum distance of 1 cm from the target or control stimulated area, a criterion used also in others study done by our group (see for instance Fornia et

al. 2018 and 2019), in order to minimize possible misleading effects due to tumor infiltration and/or dislocation (see p. 6 lines 24-25, p. 7 line 1).

Finally, please report tumor's grade, and size in relation to neurological and neuropsychological profiles for each patient.

RESPONSE

As mentioned above, we apologize for the lack of clarity about this important aspects and, also in agreement with similar concerns by Reviewer#1 and Reviewer#2, we added in the revised version a new table including the patients' details (see Table 2).

2) The approach is strongly guided by a-priori hypothesis about the role of PMC and S1. Please provide a justification for the selection of S1 (another reasonable choice would have been M1), as compared to other regions implicated in anosognosia (e.g. TPJ or insula).

RESPONSE

In the revised version, we added a new paragraph in Method section "Brain areas selection criteria" to justify our *a priori* selection of target and control areas (see p. 8-9).

3) PMC is a big place. The sites seem to be located in ventral PMC. It would be helpful to see the sites on more precise maps of PM (e.g. the Julich atlas). This is important for the interpretation. Specifically, what is the relationship of these premotor and S1 sites vs. M1 sites that produce also motor deficits? I am wondering if the PM sites were 'hand' related or perhaps more generally involved in 'action' planning of hand gestures, or some higher planning function? This is relevant since some sites appeared to be interfering with the verbal output indicating a mouth representation.

RESPONSE

We totally agree with the Reviewer#3 that PMC is a big place and, in the revised version of Figure 2, we plotted the boundaries of the PMC subparts proposed by Mayka et al (2006) template. This template is one of the few trying to anatomically separate dorsal and ventral PM based on functional metanalysis. Moreover, the subdivision proposed by Mayka is quiet in agreement also with structural data coming from Tomassini et al. 2007 and more recently from dPM characterization proposed by Genon et al. 2018. Furthermore, also according to the Reviewer#2's suggestion, we added a more detailed discussion about the functional meaning of the ventral sector of PMC where we found effective sites (p. 20 lines 7-33; p. 21 lines 1-16). Note also that, in this part of the manuscript, we incorporated the discussion of the disconnection analyses suggested below (see point 4).

4) A related approach to solve these issues would be to try to use the location of seeds, as compared to control seeds, to map functionally or structurally the networks involved. This could be done with indirect structural functional disconnection approaches as proposed by Thiebaut de Schotten (Foulon et al.) and Fox and colleagues (Boes et al.)

RESPONSE

We thank very much the Reviewer#3 for this constructive suggestion. We think that this analysis can really improve the quality of our work and we did our best to apply the suggested approach to our data. In the revised version, we described in details the new analysis we performed, including both lesion-based disconnection analysis in AHP patients and virtual disconnection analysis in our DES model of AHP (see Methods: p. 14 lines 13-26; p. 15 lines 1-7); the results of this disconnection analysis (see Results: p. 17 lines 11-31) and our interpretation (see Discussion: p. 20 lines 28-33; p. 21 lines 1-16).

5) I am skeptical that the 'negative' sites did not produce any EMG or motor effect. It would be important to see a quantitative analysis of this point. Can you report on movie or EMG data on this issue?

RESPONSE

We are not sure about what the Reviewer#3 meant for "negative" sites here. Does he mean sites inducing a motor arrest (altered HMT)? If so, it is possible to directly see the patient's behavior in Video 1-2 (in supplementary materials). The analysis of the EMG activity is reported in the Results section and plotted in Figure 2 A, B and C regarding single patients. Moreover, we also add in the bar graphs of Fig 2E the amount of RMS activity during DES of the same sites evoking HMT disruption, in rest condition (the hand and arm was completely relaxed along the arm rest). As the bar graph clearly show, the DES during the rest condition in both vPMC and S1 did not evoked involuntary muscles activation.

6) Do the authors have an hypothesis on why subjects reported correctly about 15% of the trials during PMC stimulation? Was the EMG different in those trials? Was there a threshold effect?

RESPONSE

We thank the Reviewer#3 for this comment that give us the opportunity to clarify that this percentage correspond to one trial of PMC stimulation in one patient, as evident in the new Table 2 in supplementary materials (N:4). In the same patient, at the same intensity, we found two positive effects more dorsally in which the DES disrupted both HMT and MMt. This might suggest the presence of a precise cortical spot in which DES is more effective in inducing a double simultaneous effect on motor execution and monitoring. Importantly, thanks to this comment, we added a missing data, i.e. the range of our stimulation intensity, that is between 2 and 4.5 (see p 16, line 3). The patient's stimulation intensity was in this range, thus making unlikely that a threshold effect can explain the lack of effect here.

7) A related issue is explicit vs. implicit motor awareness. Please comment whether the authors believe this distinction is purely quantitative (more or less 'intention' become verbally aware) or reflect distinct processes.

RESPONSE

The distinction between explicit and implicit anosognosia, mentioned by Reviewer#3, is very relevant in the neuropsychological context as suggested for instance by Berti et al., 1996 and Marcel et al., 2004 (see also Nardone et al., 2007). More recently Cocchini et al. (2010) in order to study implicit vs explicit aspects of motor unawareness asked their subjects to perform a series of bimanual tasks, which are usually accomplished using two hands, but could also be performed using one hand only. Although most patients with AHP tend to accomplish these tasks as if they could use both hands, some patients approached the task using one hand, thereby exhibiting some implicit knowledge of their motor deficit. The presence of such a dissociation between implicit and explicit aspects of motor awareness in AHP patients seems to suggest that this distinction reflects distinct process. However, we think that, at least in the context of intraoperative brain mapping, it should be very difficult to include a battery of ecological bimanual movements to investigate whether compensatory strategies are used by the patients during PMC stimulation, suggesting a preserved/altered implicit awareness of their motor impairment. In our revised version, we clearly underlined that our MMt is related to the explicit motor awareness (see p 18, lines 23-29).

8) Anosognosia for hemiplegia is more common after right hemisphere lesions. Do they expect to see the

same effect for right PM DES but in the left arm? Shall we then think of the neuropsychological syndrome as reflecting an impairment of a higher level monitoring system that is distinct from the contralateral motor/premotor system?

RESPONSE

This is an important point that can be investigated in future studies. Theoretically, for consistency with the neuropsychological data, we should expect to see an even greater effect for the right PMC stimulation. Indeed, as noticed by the Reviewer#3, AHP has been traditionally associated to right-brain damage, even if this disorder emerges also in left brain-damaged patients, at least when the assessment avoids language-related problems. Unfortunately, as we acknowledge in our manuscript, we were not able to test right brain tumor patients with this DES protocol due to clinical constraints (only in left brain-tumor patients neurophysiological brain-mapping is mandatory to preserve language-related functions). However, we think that our left PMC data, reproducing a virtual model of AHP in the left hemisphere, seems to confirm that there is no absolute right-brain dominance for motor awareness. With respect to the higher level monitoring system, we think that, at least in the motor context, the monitoring function is implemented in the same neural network responsible for the process that has to be controlled; i.e. in the contralateral motor/premotor system. However, to verify this hypothesis with DES, future experiments are needed, to compare the effect of both right and left PMC stimulation on the monitoring of either contralateral or ipsilateral hand movements. In the revised version, we added this consideration in our Discussion (see p. 18 lines 34: p. 19 lines 1-4).

Reviewers' Comments:

Reviewer #1:

Remarks to the Author:

[Redacted]

Reviewer #2:

Remarks to the Author:

The authors addressed all my questions. The MS is suitable for publication in Nat Comm

****REVIEWERS' COMMENTS:**

Reviewer #1 (Remarks to the Author):

[Redacted]

RESPONSE

[Redacted]

Reviewer #2 (Remarks to the Author):

The authors addressed all my questions. The MS is suitable for publication in Nat Comm

RESPONSE

We thank the Reviewer#2 for the time dedicated to our work and for her/his positive comment on our revised version.